# Chitin Nanofibril-Nanolignin Complexes as Carriers of Functional Molecules for Skin Contact Applications

**DOI:** 10.3390/nano12081295

**Published:** 2022-04-11

**Authors:** Maria-Beatrice Coltelli, Pierfrancesco Morganti, Valter Castelvetro, Andrea Lazzeri, Serena Danti, Bouchra Benjelloun-Mlayah, Alessandro Gagliardini, Alessandra Fusco, Giovanna Donnarumma

**Affiliations:** 1Consorzio Interuniversitario Nazionale per la Scienza e Tecnologia dei Materiali (INSTM), 50121 Florence, Italy; valter.castelvetro@unipi.it (V.C.); andrea.lazzeri@unipi.it (A.L.); serena.danti@unipi.it (S.D.); alessandra.fusco@unicampania.it (A.F.); giovanna.donnarumma@unicampania.it (G.D.); 2Department of Civil and Industrial Engineering, University of Pisa, 56122 Pisa, Italy; 3Academy of History of Health Care Art, 00193 Rome, Italy; pierfrancesco.morganti@iscd.it; 4Department of Chemistry and Industrial Chemistry, University of Pisa, 56124 Pisa, Italy; 5Compagnie Industrielle de la Matière Végétale (CIMV), F-31670 Labège, France; b.benjelloun@cimv.fr; 6R & D Texol, 65020 Alanno, Italy; alessandro.gagliardini@texol.it; 7Department of Experimental Medicine, University of Campania Luigi Vanvitelli, 80138 Naples, Italy

**Keywords:** chitin nanofibrils, sodium ascorbyl phosphate, vitamin C, vitamin E, lutein, nicotinamide, glycyrrhetinic acid, silver nanoparticles, skin, polysaccharide, nanolignin, keratinocytes, immunomodulation

## Abstract

Chitin nanofibrils (CN) and nanolignin (NL) were used to embed active molecules, such as vitamin E, sodium ascorbyl phosphate, lutein, nicotinamide and glycyrrhetinic acid (derived from licorice), in the design of antimicrobial, anti-inflammatory and antioxidant nanostructured chitin nanofibrils–nanolignin (CN-NL) complexes for skin contact products, thus forming CN-NL/M complexes, where M indicates the embedded functional molecule. Nano-silver was also embedded in CN-NL complexes or on chitin nanofibrils to exploit its well-known antimicrobial activity. A powdery product suitable for application was finally obtained by spray-drying the complexes co-formulated with poly(ethylene glycol). The structure and morphology of the complexes was studied using infrared spectroscopy and field emission scanning electron microscopy, while their thermal stability was investigated via thermo-gravimetry. The latter provided criteria for evaluating the suitability of the obtained complexes for subsequent demanding industrial processing, such as, for instance, incorporation into bio-based thermoplastic polymers through conventional melt extrusion. In vitro tests were carried out at different concentrations to assess skin compatibility. The obtained results provided a physical–chemical, morphological and cytocompatibility knowledge platform for the correct selection and further development of such nanomaterials, allowing them to be applied in different products. In particular, chitin nanofibrils and the CN-NL complex containing glycyrrhetinic acid can combine excellent thermal stability and skin compatibility to provide a nanostructured system potentially suitable for industrial applications.

## 1. Introduction

The increasing demand for skin contact products with high functional (antimicrobial, anti-inflammatory, antioxidant, etc.) performances, but also reduced environmental impacts, is fostering the use of renewable and, in several applicative sectors, biodegradable materials [1,2,3,4,5,6,7]. The latter may include both the matrix and vehicle of the formulation, such as, e.g., polysaccharides, and the active species introduced as co-formulants or as modifiers and, in particular, molecules extracted from biomass or agri-food waste, also exploiting nanotechnology [8,9,10,11,12,13].

Chitin nanofibrils (CNs) can be obtained from chitin (i.e., poly(β-(1→4)-N-acetyl-D-glucosamine) [14,15], which is extractable from seafood waste (crustaceans) [16], and also from mushrooms [17] and insects [18,19]. They show antimicrobial and anti-inflammatory properties that have been investigated for applications in packaging [20,21] and in cosmetics [22,23], and in biomedical applications [24,25].

Nanolignin (NL) [26,27], produced from wood [28] and other biomass sources [29,30], shows excellent antioxidant properties [29,30,31] due to its phenolic group-rich macromolecular structure. Due to its typically negative surface charge, lignin can be easily complexed with CN, which is positively charged in slightly acidic water solutions [32]. This process can be used for encapsulating useful active molecules in a nanostructured carrier [33,34,35], thus producing bionanosystems [8], referred to in the present paper as CN-NL/M.

Polymer nanoparticles with a positive surface charge, such as CNs, are capable of disturbing the lipid lamellae of stratum corneum; if loaded with molecular species intended for percutaneous administration, such nanoparticles have been found to enable better diffusion of the entrapped active ingredients through the outer skin layers [36]. On the contrary, nanoparticles with a negative surface charge seem to inhibit the diffusion of the active ingredients beyond the more external corneocytes. In the case of CN-NL complexes containing specific molecules, the modulation of this capacity can be achieved, as well as the possibility of extending the combination of properties provided by the specific bionanosystem [33]. Several naturally occurring molecules of potential interest for skin contact applications can be considered to be active ingredients to be embedded in the CN-NL complexes.

Alpha-tocopherol, the main component of the vitamin E group, is a powerful antioxidant and free radical scavenger and the most important vitamin among those responsible for protecting cell membranes against peroxidation [36,37]. Wheat germ oil, sunflower oil, rice bran oil, canola oil, and palm oil are the natural products richest in vitamin E [38]. Naturally occurring in human skin, where exposure to unfavorable environmental conditions (pollution and UV radiation) may cause its depletion, it plays a role in the protection of epidermis from oxidative stress as a photo-protectant and immunosuppression compound. This vitamin is widely used as an antioxidant in protective and anti-aging formulations in cosmetics, often together with vitamin C [39,40].

Vitamin C is a potent antioxidant compound that can prevent changes associated with photoaging [41]. As a natural compound, it is present in citrus fruits, green leafy vegetables, strawberries, papaya and broccoli; it is also one of the human antioxidant species in charge of protecting the skin from reactive oxygen species (ROS). Vitamin C is also essential for collagen synthesis [42]. Vitamin C is generally used in cosmetics as sodium ascorbyl phosphate (as in this paper) because it is more stable in the oxidative process [43].

Lutein [44] is a member of the carotenoid family and is extracted from the marigold flower, serving as an ancillary light-gathering pigment to protect organisms against the toxic effects of ultra-violet radiation and oxygen. Lutein is found in the macula of the human retina, as well as in the human crystalline lens. Up to now, lutein is used as an antioxidant compound especially for diet supplements and protective solutions for the eyes [45].

Glycyrrhetinic acid (GA), obtained from the licorice root, shows antibacterial and anti-inflammatory activity, which is mainly achieved by reducing the availability of cortisol to the adipocytes [46,47]. GA shows antibacterial activity against Staphilococcus aureus, a major pathogen in humans causing serious problems due to its antibiotic resistance. Thus, GA has been used in cosmetics to enhance the appearance of dry or damaged skin by reducing flaking and restoring suppleness [48].

Nicotinamide, the amide of B3 vitamin, is a hydrophilic natural compound with antimicrobial, antipruritic, photoprotective, sebostatic, lightening and vasoactive effects, depending on its concentration [49]. It can inhibit sirtuins, which are proteins playing critical roles in cellular responses to environmental stressors. Moreover, nicotinamide is able to modulate various cytokines and stimulate the cytokine leptin, which promotes wound healing and immune function. Finally, it enhances DNA repair following UV irradiation. For all these reasons, nicotinamide is currently used in dermatology for the treatment of various skin conditions, including acne and rosacea, and in cosmetology to treat sensitive and aged skin, exploiting its photoprotective and whitening activity [50].

Silver nanoparticles can be considered to be currently a non-bioavailable antimicrobial reference with respect to CN-NL-based complexes. The bactericidal and fungicidal activities of silver nanoparticles have resulted in their use in several consumer industry products, including cosmetic formulations [51,52,53]. The adopted technology exploits a devised system to store silver ions and to allow its controlled release. The mechanisms behind the activity of silver against bacteria are not fully elucidated yet, but it is widely accepted that the main antibacterial effect is due to its partial oxidation and release of silver ions (Ag+) in the presence of water, with their efficiency enhanced by the high surface-to-volume ratio of silver nanoparticles. The free Ag+ is capable of crossing the cell membrane, hindering the production of ATP and DNA replication; in addition, Ag+ interacts with bacterial proteins, disrupting protein synthesis. Finally, silver nanoparticles directly damage cell membranes, causing cell lysis [54]. While highly effective, studies are still required to reduce non-specific silver nanoparticles’ cytotoxicity and enhance bioavailability and stability [55].

The encapsulation of functional molecules extracted from biomass in specific nanostructured systems has been shown to be an appropriate strategy to protect them during their application, especially if a thermal treatment at high temperatures, such as in polymer extrusion, is required [8,56]. Coiai et al. [57] evidenced the efficacy of encapsulation in hydrotalcites for protecting rosmarinic acid and acetyl salicylic acid from thermal degradation upon extrusion in a polyethylene matrix. CN-NL and CN-NL incorporating a glycyrrhetinic acid complex (CN-NL/GA) are reported to be thermally stable up to about 250 °C [33]. Nevertheless, the stability of such complexes when loaded with functional molecules has not been evaluated so far. Regarding skin compatibility, the HaCaT cell line represents an established [58] model of human keratinocytes (i.e., epidermis) which can give preliminary information about molecules or complexes with respect to their inflammatory reactions through the expression of several cytokines and to antimicrobial responses through the expression of antimicrobial peptides of the innate immunity, such as β-defensin.

This paper aims to provide a systematic platform of physical–chemical, morphological and cytocompatibility comparative data for the correct selection of complexes based on CN and NL, embedding vitamin E, sodium ascorbyl phosphate, lutein, nicotinamide, and glycyrrhetinic acid, respectively, as potentially useful surface modifiers of skin contact products and items. The complexes were prepared as powdery products via spray-drying, using poly(ethylene glycol) (PEG) as an anti-agglomerating agent. Silver nanoparticles were also included in formulations with CN and CN-NL to allow the comparative evaluation of the relevant properties of the selected active natural molecules with the well-known ones of Ag nanoparticles.

## 2. Materials and Methods

### 2.1. Materials

CNs, coming from crustaceans’ waste, was prepared with a patented technology [59] using shrimp-sourced chitin. A spray-dried sample containing 2 wt% of poly(ethylene-glycol) PEG with a molecular weight Mw of 8000 (purchased from Aldrich, St. Louis, MO, USA) (CN-PEG) was prepared.

NL was provided by Compagnie Industrielle de la Matière Végétale (CIMV, Paris, France), named Biolignin™ [60]. The lignin content was 91% and the average molecular weight was 980 g/mol. The concentration of aromatic and aliphatic -OH was 1.7 ± 0.2 mmol/g and 1.9 ± 0.1 mmol/g, respectively. The concentration of carboxylic groups was 0.7 ± 0.05 mmol/g.

CN-NL and CN-NL containing the different molecules (CN-NL/M) were prepared via gelation and obtained in powder using a Buchi Mini B-190 spray-dryer (Flawil, Switzerland). [32,59,61]. Powdered samples were synthesized using spray-drying technology with 2 wt% of PEG to avoid the electrostatic aggregation of nanofibrils. Active molecules were encapsulated in the range of concentrations from 0.2% to 5% calculated on the basis of chitin weight. The ratio between CN and NL is 2:1 by weight. The content of molecules in CN-NL/M is reported in Table 1.

The CN modified with Ag nanoparticles (0.1% wt) was prepared according to a patented technique [62]. A narrow size distribution of particle diameters with an average size equal to 61.9 nm was obtained [63].

Vitamin E was obtained from Sigma Aldrich; sodium ascorbyl phosphate was purchased from BASF Care (Monheim, Germany); Lutein was purchased from Kemin (Des Moines, IA, USA); glycyrrhetinic acid was purchased from Sigma Aldrich (St. Louis, MO, USA); nicotinamide was purchased from Suzhou Greenway Biotechi (Suzhou, Jiangsu, China).

HaCaT, Dulbecco’s Minimal Essential Medium (DMEM), L-glutamine, penicillin, streptomycin, and fetal calf serum were purchased from Invitrogen (Carlsbad, CA, USA). Fetal bovine serum and AlamarBlue^®^ were purchased from Thermo Fisher Scientific (Waltham, MA, USA). Dulbecco’s phosphate-buffered saline (DPBS), silver nitrate, pyrogallol, sodium thiosulphate, nuclear fast red, dimethyl sulfoxide (DMSO), MgCl_2_, and MTT were purchased from Sigma-Aldrich (Milan, Italy). Aluminum sulphate was purchased from Carlo Erba (Milan, Italy).

### 2.2. Morphological Characterization of CN, NL, and CN-NL Complexes

The morphology of the material samples and complexes was investigated via field emission scanning electron microscopy (FESEM) by using an FEI FEG-Quanta 450 instrument (Field Electron and Ion Company, Hillsboro, OR, USA). The samples were sputtered with gold (Gold Edwards SP150B, Burgess Hill, UK) before analysis.

### 2.3. Chemical Structure and Thermal Stability Charachterisation of CN, NL, and CN-NL Complexes

The powdered samples were characterized via infrared spectroscopy using a Nicolet T380 Thermo Scientific instrument equipped with a Smart ITX ATR accessory with a diamond plate. The spectra are collected in ATR mode in a 4000–550 cm^−1^ range with a resolution of 4 (collect one point each 1.921 cm^−1^) by accumulating 256 spectra.

Thermogravimetric tests of the powders were performed on 4–10 mg samples using a Mettler-Toledo Thermogravimetric Analysis/Scanning Differential Thermal Analysis (TGA/SDTA) 851 instrument operating with nitrogen as the purge gas (60 mL/min) at a 10 °C/min heating rate in the 25–800 °C temperature range.

### 2.4. In Vitro Culture of HaCaT

Immortalized human keratinocyte (Hacat) cell lines were cultured in Dulbecco’s Modified Eagle Medium (DMEM) supplemented with 1% pen–strep, 1% glutamine, and 10% fetal calf serum at 37 °C in air and 5% CO_2_. Subsequently, cells were dispensed into 96-well plates and left to grow until full adhesion.

When the cell monolayer was reached at an appropriate confluence, substances previously solubilized in saline solution and brought to the physiological pH (c.a.7) were added at the following dilution ratios: 1:10/1:20/1:50/1:100/1:200/1:500/1:1000/1:1500/1:2000/1:4000. Treatments, conducted in triplicate, were carried out for 48 h

### 2.5. MTT Assay

To establish the optimal non-toxic concentrations to be used in subsequent treatments, HaCaT cells were seeded at a density of 1 × 10^3^/well in 96-well culture plates (n = 2). After 24 h, the cells were treated with CN, CN-NL, and CN-NL/GA at different concentrations (from 10 µg/mL to 25 ng/mL) for 24 h and then incubated with MTT (0.5 mg/mL) at 37 °C for 4 h and, subsequently, with DMSO at room temperature for 5 min. The spectrophotometric absorbance of the samples was determined by using an Ultra Multifunctional Microplate Reader (Bio-Rad, Hercules, CA, USA) at 570–655 nm [64].

### 2.6. Anti-Inflammatory and Immune Responses Evaluation of HaCat Cells

HaCaT cells, seeded in 6-well culture plates (n = 3) until reaching 80% confluence, were treated with CN, CN-NL, and CN-NL/M at selected concentrations (i.e., 10 µg/mL, 0.2 µg/mL, and 0.5 µg/mL, respectively) for 6 h and 24 h. At the end of the experiment, total ribonucleic acid (RNA) was isolated and one microgram of this were reverse-transcribed into complementary deoxyribonucleic acid (cDNA) using random hexamer primers (Promega, Milan, Italy) at 42 °C for 45 min, according to the manufacturer’s instructions. The anti-inflammatory and immune responses of HaCaT cells were evaluated by assaying the expression of pro-inflammatory cytokines IL-1α, IL-1β, IL-6, IL-8, and TNF-α, anti-inflammatory cytokine TGF-β, and antimicrobial peptide HBD-2 via real-time PCR with the LC Fast Start DNA Master SYBR Green kit from Roche Applied Science (Euroclone S.p.A., Pero, Italy) using 2 µL cDNA, corresponding to 10 ng of total RNA in a 20 Mm µL final volume, 3 mM magnesium chloride (MgCl_2_), and 0.5 µM of sense primer and antisense primers (Table 2).

At the end of each run, the melting curve profiles were obtained by cooling the sample to 65° C for 15 s and then heating it slowly at 0.20 °C/s up to 95 °C, with continuous measurement of fluorescence to confirm the amplification of specific transcripts. Cycle-to-cycle fluorescence emission readings were monitored and analyzed using LightCycler^®^ software (Roche Diagnostics GmbH, Monza, Italy). Melting curves were generated after each run to confirm the amplification of specific transcripts. We used the b-actin coding gene, one of the most commonly used housekeeping genes, as an internal control gene. All reactions were carried out in triplicate, and the relative expression of a specific mRNA was determined by calculating the fold change relative to the b-actin control. The fold change of the tested gene mRNA was obtained with LightCycler^®^ Software 3.5.by using the amplification efficiency of each primer, as calculated using the dilution curve. The specificity of the amplification products was verified by subjecting the amplification products to electrophoresis on 1.5% agarose gel and visualization using ethidium bromide staining [65].

## 3. Results and Discussion

### 3.1. Structure, Thermal Stability, and Morphology of CN-NL Spray-Dried Complexes

A CN suspension in slightly acidic water, where nanofibrils are positively charged, was prepared in the presence of the active molecules, followed by the addition of negatively charged NL to obtain the corresponding CN-NL complexes. Powdered samples were, thus, obtained by spray-drying CN added with some PEG to avoid the agglomeration of nanofibrils, as already observed in a previous work [66]. Active molecules were encapsulated in the range of concentrations from 0.2% to 5%, calculated on the basis of chitin weight.

Chitin occurs in nature as crystalline microfibrils forming structural components in the exoskeleton of arthropods or in the cell walls of fungi and yeasts. Depending on its source, chitin occurs as three allomorphs, namely the α, β, and γ forms. In both α and β structures, the chitin chains are organized in sheets tightly held together by intra-sheet hydrogen bonds. This tight network, dominated by hydrogen bonds, maintains the chains very close to each other (Figure 2a) [67]. In α-chitin, there are also some inter-sheet hydrogen bonds involving the association of the hydroxymethyl groups of adjacent chains [68].

In the FTIR spectrum of pure CN (see Figure 2b), the bands at 3443 cm^−1^, 3261 cm^−1^, and 3109 cm^−1^ are attributed to the stretching vibration of intramolecular hydrogen-bonded OH(1) in the C-OH···O=C structure (red circle in Figure 2a), the NH stretching restricted by C-NH···O=C hydrogen bonds (blue circle), and the OH(2) stretching restricted by intermolecular C-OH···H–O–C hydrogen bonds (green circle), respectively [69,70]. The shape and intensity of these peaks will change if the hydrogen bonding network in chitin is altered. The bands ranging from 2876 to 2961 cm^−1^ represent symmetric CH and CH_3_ and asymmetric CH_2_ and CH_3_ stretching. The CH bending, symmetric CH_3_ deformation, and CH_2_ wagging bands appear at 1376 and 1316 cm^−1^. The splitting of the amide I vibration at 1656 and 1620 cm^−1^, assigned to the stretching of C=O groups hydrogen-bonded to the N–H groups of the intra-sheet chain and to the hydroxymethyl group of the next chitin residue of the same chain, correspond to the spectrum of α-chitin [71], although the band at 1621 cm^−1^ has also been associated with either a combination band or an enol form of the amide moiety [72]. The absorption bands at 1558 and 1312 cm^−1^ correspond to amide II (N-H bending) and amide III (C-N stretching), respectively [73]. The bands ranging from 1071 to 1000 cm^−1^ are attributed to the asymmetric C-O-C and C–O stretching [74].

In the IR spectrum of the spray-dried CN treated with PEG (red in Figure 2c), the presence of PEG cannot be easily detected. However, the characteristic bands of pure PEG at 1340, 1070, and 960 cm^−1^ may account for the variations in the relative peak intensities in the 1350–950 cm^−1^ range in the spectra of CN and CN-NL with and without PEG (see Figure 2), in agreement with the literature [75]. The peaks at 3100–3440 cm^−1^, influenced by hydrogen bonds, were not significantly altered by the presence of PEG and NL.

Lignin is an amorphous, polyphenolic biomaterial resulting from an enzyme-mediated dehydrogenative polymerization of the three phenylpropanoid monomers coniferyl, synapyl, and p-coumaryl alcohol (Figure 2d). The spectrum of NL used in this work (Figure 2e) was examined, and similar peaks were found in the CN-NL spectrum. The slight shift of peaks is attributable to the interactions between CN and NL. This CN-NL spectrum (Figure 2c) presented the characteristic strong and broad symmetric and asymmetric –CH_2_ stretching band at 2881 cm^−1^, a very weak carbonyl stretching band at 1735 cm^−1^, and the absorption at 1465 cm^− 1^ due to the C–H asymmetric deformation in the methyl, methylene, and methoxy groups of aromatic rings, in agreement with the literature [76]. On the other hand, the generally weak absorption at about 1400 cm^−1^ expected from the aromatic C-C stretching and the C-H in-plane deformation could not be detected. Additional absorptions typical of the spectra of lignin are a weak band at 1340–1345 cm^−1^ originating from phenolic O-H and aliphatic C–H in methyl groups. In the –C-O-C and alcoholic C-O stretching bands from 1000 to 1260 cm^−1^, the band ranging from 1050 to 1150 cm^−1^ represents the C-O-C stretching of the ether group along the polymer chains. The CH bending, symmetric CH_3_ deformation, and CH_2_ wagging bands, appearing at about 1376 and 1318 cm^−1^, can be attributed to chitin molecules. The peak at 840 cm^−1^ is typical for aromatic C–H out-of-plane bending in C2 and C6 of syringyl units (Sinapyl alcohol fraction).

The results of the thermogravimetric analyses performed on pure CN, pure NL, PEG, spray-dried CN (with PEG), and spray-dried CN-NL (with PEG) powders are shown in Figure 3, while some characteristic data drawn from the recorded thermograms are listed in Table 3.

The presence of just a small 2 wt% of PEG significantly influenced the thermo-degradation behavior. Pure CN resulted in being more thermally stable than the complexes including PEG and NL (bold black and green curves). All samples showed substantial thermal stability below 140 °C, the small weight loss observed below that temperature being likely due to residual water. Above this temperature and up to 200 °C, the thermograms of CN and of its spray-dried complexes showed the onset of a first degradation step with associated small (lower than 5 wt%) weight loss, followed by a sharp second degradation step with a maximum degradation rate at about 300 °C and with about 60–70% weight loss in the 200–350 °C range for the two complexes, corresponding to the less sharp and comparatively lower weight loss observed for CN (about 20% weight loss) and NL. Given the high stability of PEG, the worsened stability of spray-dried powders is not easy to explain. The different morphology of the samples may play a significant role.

Pure CN 20% suspension in water was diluted 1:1000 in volume and deposited on a glass support to observe the morphology of the CNs. The spray-dried powder of CN in the presence of a few percent of PEG and also the complex CN-NL obtained in the presence of some PEG were suspended in deionized water (1:1000 in volume) and deposited on glass supports. Then, the three samples were characterized using field emission SEM. Pure NL powder was also characterized.

The dimensions of CN are on a nano-scale (Figure 4a,b) and the shape is needle-like. The powder obtained by drying the CN was also characterized and consisted of scraps of agglomerated CN (Figure 4c). In fact, the CNs assemble easily because of strong inter-fibrillar interactions. According to that, the surface area of dried CN powder determined via BET (Micromeritics Gemini V) measurements was found to range from 1.7 to 3.4 m^2^/g. The surface area is small and in agreement with micrometric dimensions. It means that CNs recovered from aqueous suspensions by drying easily to form resistant bundles of micrometric dimension.

The morphology was completely different for CNs obtained via spray-drying in the presence of a few percent of PEG (Figure 4d). Round micrometric particles can be observed in both powder (Figure 4d) and diluted suspensions (Figure 4e), indicating the significant stability of these micrometric particles. The shape is probably attributable to the self-assembly of fibrils in the presence of small PEG domains, behaving like a glue, during spray-drying. It is interesting to notice the surface texture (Figure 4f) where cavities due to the removal of water during spray-drying can be observed, improving the surface-to-volume ratio and, thus, reasonably increasing the activity of CN with respect to the pure dried CN.

Pure lignin powder (Figure 4i) consists of big micrometric particles, but at higher magnifications a nano-structured system consisting of smaller particles can be revealed (Figure 4i). Hence, after stirring in slightly salted water, this lignin, which tends to accumulate negative charges, generates nanoparticles. In the gelation process used to obtain CN-NL complexes, both CNs (with 2wt% of PEG) and NL are suspended in water, and then they are spray-dried. The resulting shape of CN-NL complexes (CN-NL-PEG) is shown in Figure 4f. Micrometric round particles were formed. Furthermore, when the particles are suspended in water and deposited on a surface, ellipsoidal platelets (Figure 4g) are formed in which chitin nano-fibrils are assembled both with NL and PEG (Figure 4h), resulting in flat micrometric particles showing a smooth surface. These micro platelets can, thus, adhere well on a surface thanks to their flat shape, forming micrometric agents for the functionalization of surfaces exploiting nanotechnology. Thus, the presence of lignin changed the shape that CNs can assume completely, granting the maximum improvement of the surface-to-volume ratio when deposited from water suspension.

### 3.2. Structure and Thermal Stability of CN-NL/M

In Figure 5 and Figure 6, the comparison of the FT-IR spectra of CN-NL/M with a reference FT-IR spectrum (CN-NL) is shown. The absorption bands of the different samples are completely overlapped to the reference bands of CN-NL, as evidenced by the black lines. On the other hand, the content of active molecules was low; hence, their characteristic peaks can generate only very weak signals.

The spectra recorded for pure glycyrrhetinic acid and vitamin E evidenced the presence of peaks (Figure 5) that are attributable to their characteristic functional groups [77,78] that were found to be compatible with the related complexes’ spectra.

FT-IR spectra of CN-NL/M with three different encapsulated active molecules are shown in Figure 6. These molecules possess functional groups that overlap with those of chitin, lignin, and PEG bands. The bands ranging from 3500 to 2880 cm^−1^ are the same in all three complexes, with strong broad bands representing O-H (3360 cm^−1^), N-H (3260 cm^−1^), and C-H, CH_2_ (2880 cm^−1^) stretching vibrations. Additionally, characteristic bands of the carbonyl group of amide group are shown in all samples because they are disturbed by the presence of chitin residue. Nevertheless, active molecules have been identified through the characteristic bands in the fingerprint area of the spectra. The band ranging from 1670 to 1590 cm^−1^ is typical for double bonds in C=C, C=N, and C=O that are present in all samples. In particular, peculiar bands peaks were identified. Infrared spectra were recorded also for pure active compounds (the compounds used for the synthesis of complexes): lignin, sodium ascorbyl phosphate, nicotinamide and lutein. Specific papers related to the infrared characterization of sodium ascorbyl phosphate [79], nicotinamide [80], and lutein [81] were considered for their interpretation. It was found that the peculiar bands’ peaks identified on the spectra of complexes can be reasonably attributed to the active molecules (Figure 6).

The spectrum of the CN containing silver was recorded and compared with the one obtained by Wijesena et al. [82]. In good agreement with this paper, only some slight shifts of the C=O and C-O-C stretching bands were observed when comparing chitin to its complex with silver nanoparticles. The spectrum of the CN-NL containing silver was compared with the one of CN/AG, and it was evident that the differences in spectra are attributable to the presence of lignin, with its typical peaks (Figure 7) at 1735 cm^−1^, 1600 cm^−1^, and 1400 cm^−1^.

A TGA analysis of the CN-NL/M complexes was carried out to study their thermal stability, as these complexes may require high-temperature processing, such as, e.g., melt blending [4,66] or extrusion coating [83], for their proper application. The thermograms of CN-NL/M powder samples containing the different active molecules are reported in Figure 8.

Using data obtained from the CN-NL/M thermograms, several characteristic indicators were calculated as reported in Table 4, with the aim of comparing the stability of the different complexes. It is evident that the presence of glycyrrhetinic acid, lutein, and vitamin E slightly improved the thermal stability of CN-NL, reducing the second onset temperature, which was attributable to the biopolymeric structure degradation. On the other hand, sodium ascorbyl phosphate and nicotinamide induced some changes in the thermo-degradation behavior with respect to CN-NL. In particular, the comparatively higher amount of residue at 600 °C and the absence of the small degradation step below 200 °C in CN-NL/VC (we are assuming, here, that the higher weight loss at that temperature was the result of the release of absorbed water) may be ascribed to the well-known antioxidant properties of vitamin C; on the contrary, in the presence of nicotinamide, the onset of the first stage occurred at a slightly higher temperature (about 130 °C vs. 110 °C for CN-NL), but with a significantly higher weight loss before the following main degradation step.

In order to understand if the embedded active molecule plays a role in the decreased thermal stability of the embedding CN-NL complex, the TGA characterization was extended to the pure molecules. Their respective TGA thermograms are reported in Figure 9, along with the thermogram of CN/AG, the latter to be compared with that of the CN-NL/AG data shown in Figure 8. The onset temperature of the first observed mass loss, the inflection point and mass loss (step) of the main peak, and the final residue values are reported in Table 4. The final residue was negligible for GA, NI, LU, and VE, whereas a high amount of residue was obtained for VC (58.5 wt%). Indeed the latter, in its acid form, is known to undergo a main decarboxylation and dehydration step between 250 and 300 °C, leaving about 25 wt% of charred residue at 800 °C 99; in our case, the 10% weight loss below 200 °C and the even higher amount of residue at 800 °C are reasonably related to the sodium salt form of VC, which may release hydration water molecules at low temperatures and retain additional organic sodium salt or sodium oxide in the final residue [84]. The CN/AG resulted in a 12.01% residue, higher than that of CN-NL/AG (4.41%). Hence, the presence of NL, which is less thermally stable than CN, contributed either directly, through the generation of reactive degradation products, or indirectly, through the formation of a more porous morphology, to a more effective generation of volatile species upon thermal degradation.

The temperature of the first 5% mass loss, 5%ML, was found to be in the order:VC (70 °C) **<** NI (140 °C) **<** VE **=** LU (180 °C) **<** GA **=** CN/AG (240 °C)

Thus, VC, because of its loss of water, resulted the less thermally stable active agent, whereas GA delivered the most thermally stable molecule. The CN/AG sample resulted in being highly thermally stable, with the 5%ML temperature higher than that of CN-NL/AG, confirming again that NL reduces the thermal stability of CN, and much higher than that of CN (Table 5, pure CN powder complex with PEG). Indeed, the incorporation of silver nanoparticles has been reported to result in increased thermal stability for different polymer matrices [84,85,86]. In our case, the better thermal stability may be attributed to the presence of strong interactions between Ag and CN through the hydroxyl and carbonyl moieties of the CN structure [85].

### 3.3. HaCaT Cell Viability, and Anti-Inflammatory and Immune Responses of HaCaT Cells

The HaCaT cell line represents an established model of human keratinocytes, which can give preliminary information about the inflammatory reactions of molecules or complexes through the expression of several cytokines. The MTT viability test was carried out in order to establish the conditions to work at not-cytotoxic concentrations for each substance (Table 6) during subsequent experiments. On the basis of the obtained values, the concentrations were chosen for the highest viability, and they are indicated in Table 6 with an asterisk. In particular, it is evident that some compounds needed to be highly diluted in order to be cytocompatible with HaCat cells. This is the case of CN-NL/AG, CN-NL/VE, and CN-NL/LU, needing to be diluted at least 2000 times, with CN-NL/AG showing a viability percentage of only 62.5% ± 5%, even at a 1:4000 dilution ratio. Viability values above 100%, namely, an increment in cell viability, were obtained using CN-NL, CN-NL/VC, CN-NL/GA, and CN at optimal concentrations, namely, diluted at 1:500, 1:500, or even less, such as, 1:200 and 1:10, respectively.

Cytokines are biological molecules that act as soluble mediators of natural immunity and the immune response [86]. Among the pro-inflammatory cytokines, we investigated a panel of interleukins (ILs) and factors: IL-1, according to its two forms, α and β, because it promotes local inflammation, increases the expression of adhesion molecules and induces the proliferation and differentiation of immune cells [87]; tumor necrosis factor-alpha (TNF-α), which is a powerful mediator in inflammation, since, with a cascade of events, it is able to recruit inflammatory cells and immunoglobulins, complements, and also acts on, platelet adhesiveness, favoring the formation of thrombus [88]; IL-6 mediates the autocrine, paracrine, and endocrine involved in inflammation [89]; IL-8, which recruits and activates polymorphonuclear leukocytes, is involved in basophil chemotaxis and angiogenesis [90]. The family of β-defensins is composed of small cationic peptides produced by epithelial cells, among others, and, inherently or in response to microorganisms or cytokines, it contributes to broad-spectrum innate immunity. In particular, human β-defensin-2 (hBD-2) is an antimicrobial peptide acting as an endogenous antibiotic in the defense against Gram-positive and Gram-negative bacteria, fungi, and the envelope of some viruses, and it is involved in the innate immune response because its release is induced by proinflammatory cytokines, endogenous stimuli, infections, or wounds [91,92,93,94]. The results of cytokine expression analysis showed that the substances were endowed with an immunomodulatory activity; in fact, they were able to upregulate the expression of IL-6, TNF-α, and IL-1α and to downregulate the expression of IL-8 after 6 h. In addition, all the substances, except for CN-NL/GA, showed indirect antimicrobial activity, as they induced the expression of HBD-2 by keratinocytes after 6 h (Figure 10). All the cytokines that resulted were downregulated after 24 h, suggesting that the pro-inflammatory effect (fundamental to initiate the healing process) was actually modulated in a short time, bringing the cells back to or below their basal expression condition.

The obtained results provided useful information to properly use the CN-NL/M complexes and to select proper techniques to use them to modify the surfaces of substrates, with the aim of adding desired functional properties to skin contact products. Considering this objective, Figure 11 presents the ranking of cytocompatibility and thermal stability of the different complexes.

CN/AG and CN-NL/AG were compared with other samples in this work, as currently they are considered to be high-performance and commercially available antimicrobial nano-structured treatments. The presence of CN and CN-NL can also add to these properties additional antimicrobial and antioxidant/antimicrobial properties, respectively. The obtained results indicated that nano-silver-based systems are also thermally stable; hence, they can be applied on surfaces by exploiting a range of techniques, including high-temperature processes. Nevertheless, the compatibility with cells was not very good because of their cytotoxicity. Hence, they behave as many other nano-metal-based antimicrobials, being detrimental for both microbes and human cells. On the contrary, CN-NL/M complexes and CN can be promising antimicrobials through an indirect antimicrobial action that is based on the production of defensins from skin cells.

The selection of the best complexes for different applications should be exercised while taking into account both the knowledge about the properties of these molecules and the results of tests described in the present work. For sanitary applications, where antiseptic and anti-inflammatory properties are required, the best option consists of the use of CNs or the complex CN-NL incorporating glycyrrhetinic acid (CN-NL/GA) [33]. For cosmetics, where antioxidant and anti-inflammatory properties are desired, CN-NL or CN are the best options, although options with vitamin E [34] or glycyrrhetinic acid [35] can also be promising, taking into account specific concentrations and treatment conditions. In addition, from the point of view of the ability to modulate the expression of pro-inflammatory cytokines, our data show that the most suitable materials are CN-NL with vitamin E (CN-NL/VE). For wound dressing, where anti-inflammatory and antiseptic properties are the most required, CNs, CN-NL with glycyrrhetinic acid (CN-NL/GA) [33], and CN-NL are the best options, since they also show proliferative activity, which is very important during the tissue regeneration process, while maintaining an immunomodulatory ability. Concerning the ability to upregulate the expression of HBD-2 and, therefore, to possess indirect antibacterial activity, the most efficient agent is CN. Hence, the use of CN results in the best option if only an antimicrobial modification is required.

## 4. Conclusions

Spray-dried powders consisting of CNs or complexes of CN-NL were prepared in the present work. The spectroscopic, thermal, and morphological characterization allowed the peculiarities resulting from different compositions and preparation techniques to be evidenced. In particular, the different morphologies observed between pure CN and CN obtained via spray-drying with PEG (2 wt%) accounts for their different thermal stability. Indeed, the latter formed round and highly porous microparticles with an extended surface-to-volume ratio, resulting in a slightly decreased thermal stability with respect to the more compact CN, as shown via thermogravimetric analysis. Spray-dried CN-NL powders consisted of round microparticles, becoming flat and ellipsoidal if deposited on surfaces from water-based formulations. Due to the thermal stability of NL being lower than that of CN, the CN-NL complexes resulted in being less thermally stable than CN. The thermal stability of both CN and CN-NL was generally increased upon the incorporation of silver nanoparticles, possibly thanks to the interactions between silver nanoparticles and CN.

The compatibility with keratinocytes indicated that CN and CN-NL/GA were the functional agents that were more compatible with skin cells, being able to impart antimicrobial and antimicrobial/antioxidant/anti-inflammatory properties, respectively. Among the explored complexes, the most effective composition in terms of the ability to modulate the expression of pro-inflammatory cytokines (immunomodulatory activity) was found to be CN-NL incorporating vitamin E (CN-NL/VE).

The present paper compared several renewable nanostructured materials that could be applied to different surfaces and impart functional properties. In particular, bio-based products, such as films, non-woven materials, or tissues, can be modified by using these nanostructured agents, thus making products more environmentally and health-friendly. The obtained results provided the necessary information to properly use the CN-NL/M complexes and select proper techniques to use them to modify the surfaces of substrates to add the desired functional properties to products that are designed to stay in contact with skin, such as sanitary pads, protective items, cosmetics, and wound dressings.

## Figures and Tables

**Figure 1 nanomaterials-12-01295-f001:**
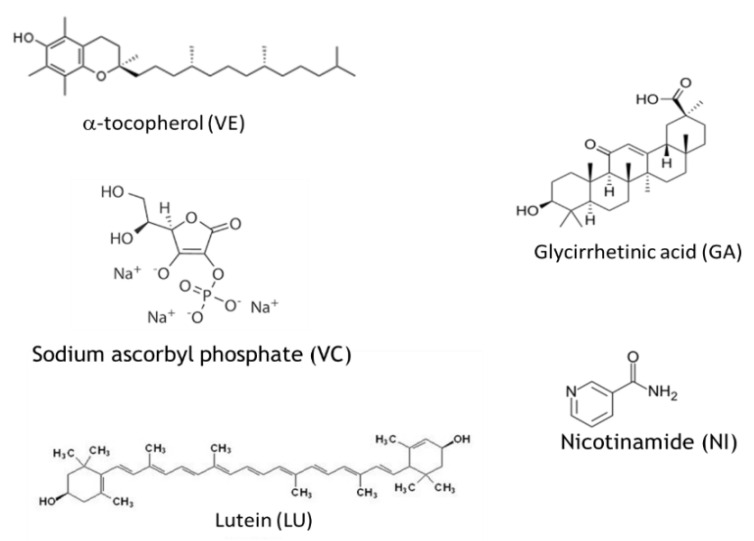
Molecular formula of the molecules encapsulated in CN-NL complexes. The abbreviations used in this paper are reported in brackets.

**Figure 2 nanomaterials-12-01295-f002:**
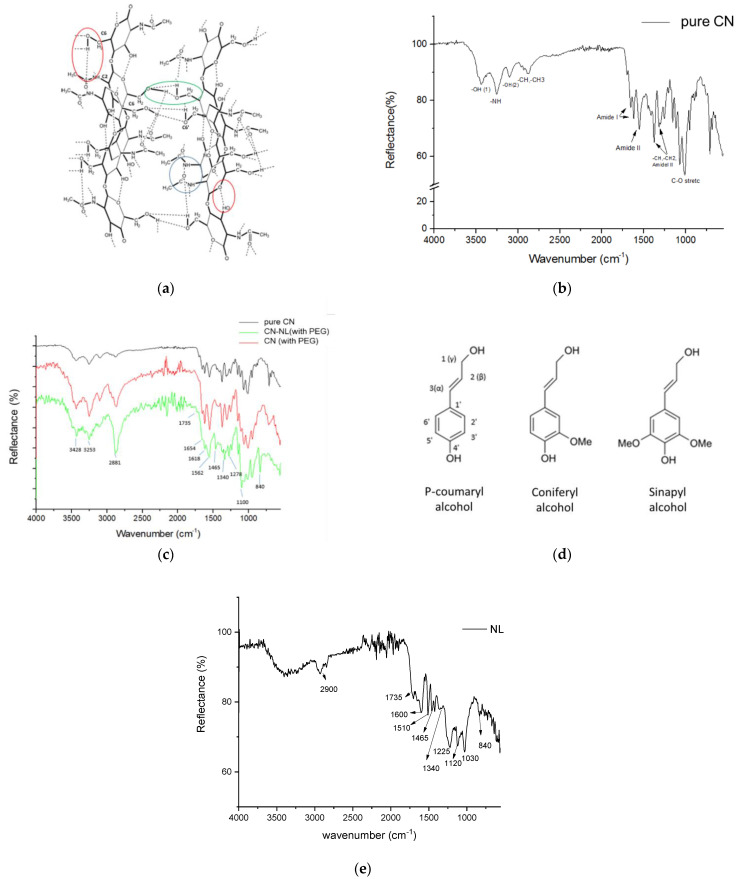
Structure and ATR infrared characterization of CN: (**a**) α-chitin molecular structure and hydrogen bonding. Colored circles represent the different hydrogen bonding processes inside the chitin crystal structure. Readapted from Pillai et al. [67], courtesy of Elsevier; (**b**) spectrum of pure CN; (**c**) spectra of pure CN (black), spray-dried CN containing PEG (red), and CN-NL (green); (**d**) main molecular fragments of lignin structure; (**e**) spectrum of nanolignin (NL).

**Figure 3 nanomaterials-12-01295-f003:**
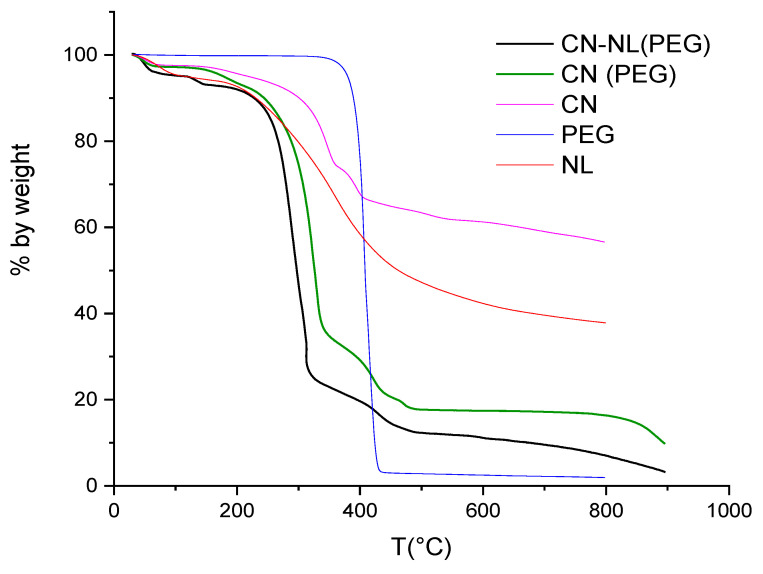
TGA thermograms of: pure CN; pure NL; spray-dried CN (with PEG), bold; spray-dried CN-NL (with PEG), bold; PEG.

**Figure 4 nanomaterials-12-01295-f004:**
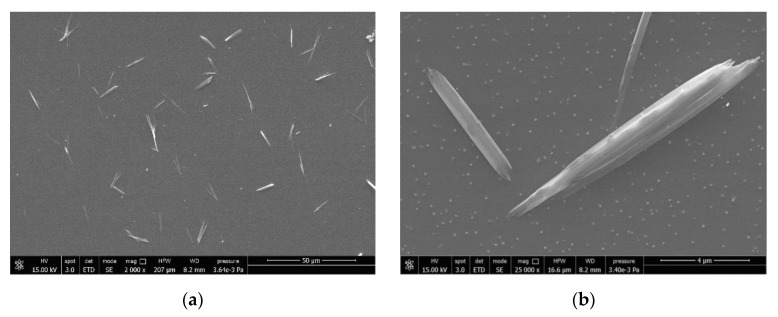
FESEM characterization of CN, NL, and CN-NL samples: (**a**) pure CN deposited from a water suspension; (**b**) magnification of pure CN deposited from a water suspension; (**c**) pure CN scraps obtained by drying CN in a water suspension; (**d**) CN particles obtained via spray-drying using PEG; (**e**) magnification of the CN particles surface obtained by spray-drying CN particles using PEG; (**f**) CN-NL particle obtained via spray-drying with PEG; (**g**) CN-NL particles obtained after suspending spray-dried CN-NL particles and depositing them on a surface; (**h**) same as (**g**), but at a higher magnification; (**i**) microparticles of lignin; (**j**) nanostructure of lignin particles.

**Figure 5 nanomaterials-12-01295-f005:**
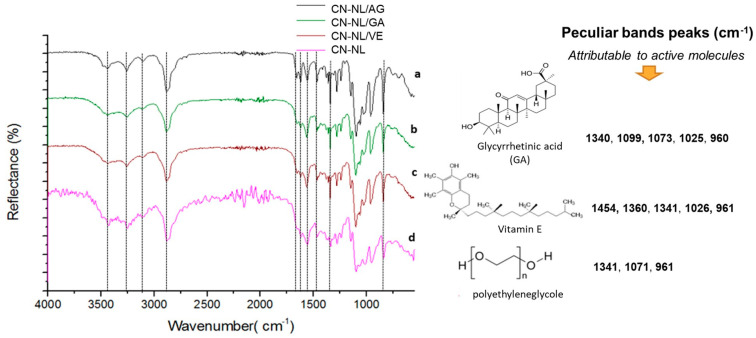
ATR infrared characterization of spray-dried CN-NL/M complexes: (a) CN-NL/AG; (b) CN-NL/GA; (c) CN-NL/VE; (d) CN-NL. The peculiar band peak values found in the spectra and attributable to VC, NI, and LU molecule are reported on the right.

**Figure 6 nanomaterials-12-01295-f006:**
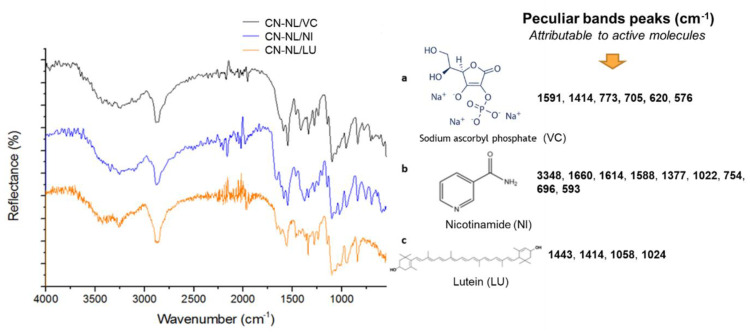
ATR infrared characterization of spray-dried CN-NL/M complexes: (**a**) CN-NL/VC; (**b**) CN-NL/NI; (**c**) CN-NL/LU. The peculiar band peak values found in the spectra and attributable to VC, NI, and LU molecule are reported on the right.

**Figure 7 nanomaterials-12-01295-f007:**
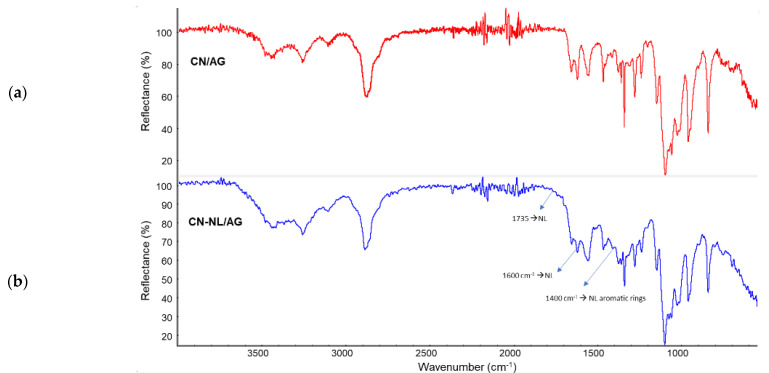
ATR infrared characterization of: (**a**) CN/AG; (**b**) CN-NL/AG.

**Figure 8 nanomaterials-12-01295-f008:**
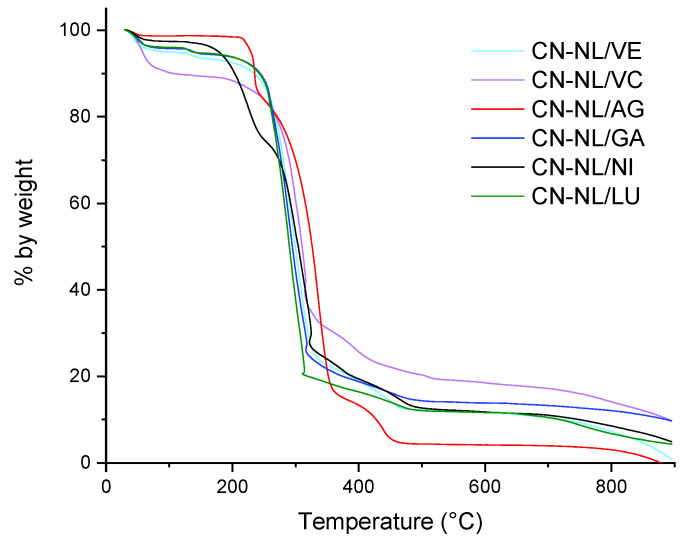
TGA thermograms of spray-dried CN-NL/M complexes.

**Figure 9 nanomaterials-12-01295-f009:**
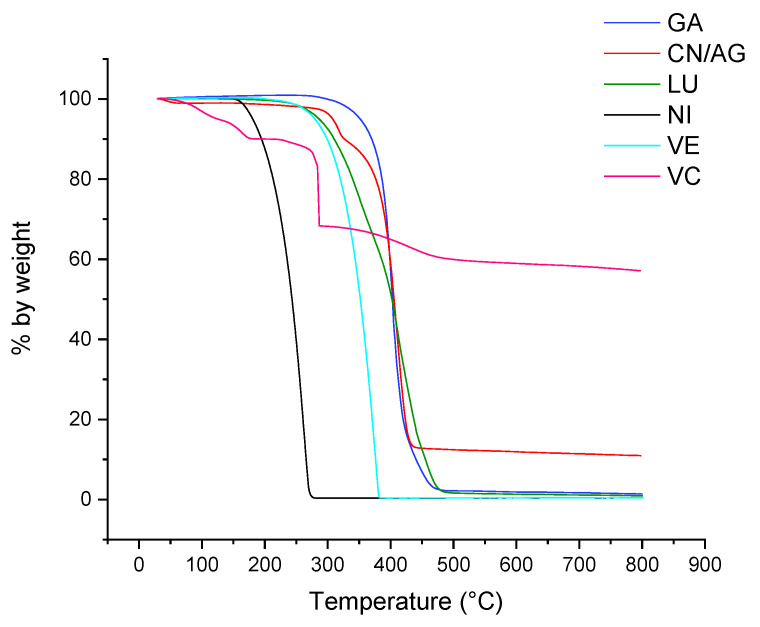
TGA thermograms of CN/Ag and of pure molecules: glycyrrhetinic acid (GA), vitamin E (VE), sodium ascorbyl phosphate (VC), nicotinamide (NI), and lutein (LU).

**Figure 10 nanomaterials-12-01295-f010:**
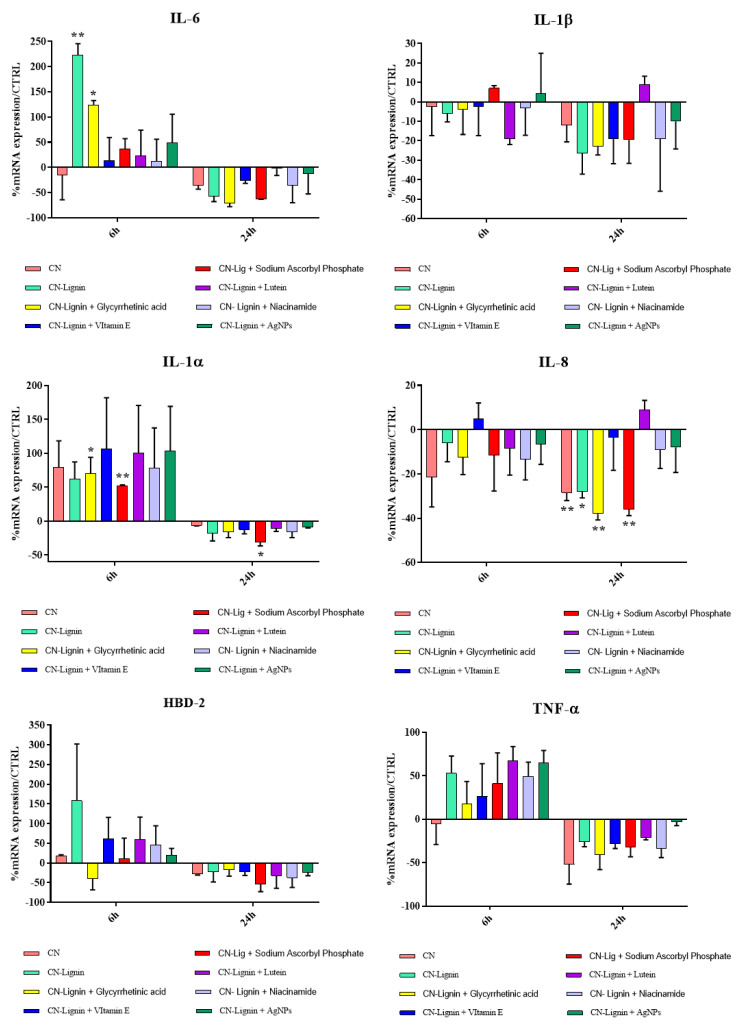
Relative gene expression of HBD-2 and proinflammatory cytokines in Hacat cells. Data are mean ± SD and are expressed as a percentage of the relative mRNAs in each group compared to unstimulated cells (CTRL). Significant differences are indicated by * *p* < 0.05 and ** *p* < 0.01.

**Figure 11 nanomaterials-12-01295-f011:**
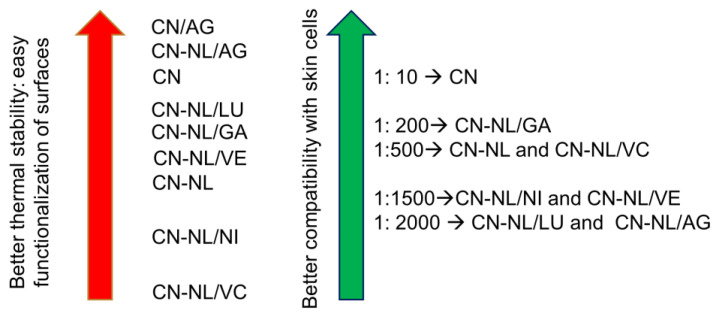
Scheme summarizing the results of tests regarding thermal stability and skin cell compatibility.

**Table 1 nanomaterials-12-01295-t001:** Powdered samples’ compositions.

Name	Sample Description ^1^
CN	Chitin nanofibrils
CN-PEG	CN with PEG8000 (2 wt%)
CN-NL	CN-NL complex with PEG8000 (2 wt%)
CN-NL/GA	CN-NL + glycyrrhetinic acid (0.2 wt%)
CN-NL/VE	CN-NL + Vitamin E (0.2 wt%)
CN-NL/VC	CN-NL + Sodium ascorbyl phosphate (0.2 wt%)
CN-NL/LU	CN-NL + lutein (0.5 wt%)
CN-NL/NI	CN-NL + nicotinamide (5 wt%)
CN-NL/AG	CN-NL + silver nanoparticles (0.1 wt%)
CN/AG	CN + silver nanoparticles (0.1 wt%)

^1^ The molecules’ structure is reported in Figure 1.

**Table 2 nanomaterials-12-01295-t002:** Primer sequences and RT-PCR conditions.

Gene	Primers Sequence	Conditions	Product Size (bp)
IL-1 α	5′-CATGTCAAATTTCACTGCTTCATCC-3′ 5′-GTCTCTGAATCAGAAATCCTTCTATC-3′	5″ at 95 °C, 8″ at 55 °C, 17″ at 72 °C for 45 cycles	421
TNF-α	5′-CAGAGGGAAGAGTTCCCCAG-3′ 5′-CCTTGGTCTGGTAGGAGACG-3′	5″ at 95 °C, 6″ at 57 °C, 13″ at 72 °C for 40 cycles	324
IL-6	5′-ATGAACTCCTTCTCCACAAGCGC-3′ 5′-GAAGAGCCCTCAGGCTGGACTG-3′	5″ at 95 °C, 13″ at 56 °C, 25″ at 72 °C for 40 cycles	628
IL-8	5-ATGACTTCCAAGCTGGCCGTG-3′ 5-TGAATTCTCAGCCCTCTTCAAAAACTTCTC	5′’ at 94 °C, 6″ at 55 °C, 12″ at 72 °C for 40 cycles	297
TGF-β	5′-CCGACTACTACGCCAAGGAGGTCAC-3′ 5′-AGGCCGGTTCATGCCATGAATGGTG-3′	5″ at 94 °C, 9″ at 60 °C, 18″ at 72 °C for 40 cycles	439
IL-1 β	5′-GCATCCAGCTACGAATCTCC-3′ 5′-CCACATTCAGCACAGGACTC-3′	5″ at 95 °C, 14″ at 58 °C, 28″ at 72 °C for 40 cycles	708

**Table 3 nanomaterials-12-01295-t003:** Inflection points (T_infl_) and associated mass variations (Δwt%), temperature at which 5 wt% mass loss is reached (5%ML), and final residue at 900 °C (R_900_), as determined from the thermogravimetric curves reported in Figure 3.

Pure CN		NL		PEG		CN (PEG)		CN-NL (PEG)	
T_infl_ (°C)	Δwt%	T_infl_ (°C)	Δwt%	T_infl_ (°C)	Δwt%	T_infl_ (°C)	Δwt%	T_infl_ (°C)	Δwt%
51.7 191.2 349.1 394.2 514.4	−2.4 −2.6 −21.2 −9.5 −2.7	72.2 361.149	−5.26 −55.63	406.2	−97.0	46.5 188.6 329.8 421.6 471.3	−1.9 −5.3 −57.5 −17.8	49.7 130.5 292.5 430	−5.3 −2.6 −69.2 −10.4
R_900_ = 58.1 wt%		R_900_ = 38.9 wt%		R_900_ = 2.8 wt%		R_900_ = 17.4 wt%		R_900_ = 12.4 wt%	
5%ML: 140 °C		5%ML: 140 °C		5%ML: 320 °C		5%ML: 140 °C		5%ML: 100 °C	

**Table 4 nanomaterials-12-01295-t004:** Analysis of thermogravimetric curves of pure molecules and CN/Ag.

Samples	5%ML (°C)	Inflection Point (Main Loss) (°C) (wt%)	Step (Main Loss) (%)	Residue (wt%)
CN/AG	240	413.6	86.4	12.01
GA	240	404.1	98.9	1.97
VE	180	367.4	99.8	0.4
VC	70	279.1	41.5	58.5
NI	140	266.7	99.7	0.3
LU	180	416.3	98.6	1.4

**Table 5 nanomaterials-12-01295-t005:** Analysis of thermogravimetric curves of CN-NL/M complexes reported in Figure 8. Spray-dried CN and CN-NL are also reported for comparison.

Samples ^(a)^	m (wt%)	5%ML	Onset 1 (°C)	Step 1 (wt%)	Onset 2 (°C)	Step 2 (wt%)	Onset 3 (°C)	Step 3 (wt%)	Residue (wt%)
CN (PEG)	-	140	159	−5.35	300	−57.54	387	−17.85	17.42
CN-NL (wt ratio 2:1)	-	100	114	−2.599	266	−69.25	406	−10.39	12.37
CN-NL/LU	0.5	110	123	−1.637	253	−73.74	387	−8.197	11.78
CN-NL/NI	5	140	189	−23.6	276	−48.58	438	−13.99	11.80
CN-NL/VC	0.5	50	80	−10	267	−56.45	373	−14.02	18.91
CN-NL/AG	0.2	200	217	−14.88	292	−68.99	416	−10.37	4.41
CN-NL/GA	0.2	100	127	−1.92	251	−68.53	472	−11.63	13.84
CN-NL/VE	0.2	100	125	−2.545	264	−70.99	435	−10.90	11.56

^(a)^ All CN-NL matrices also containing 2 wt% PEG.

**Table 6 nanomaterials-12-01295-t006:** O.D. values of viable cells after treatment with each substance at the various dilution ratios, with the respect to untreated cells, in which the O.D. is 1.15.

	CN-LIG + AgNPs	CN-LIG + Vit.E 0.2%	CN-LIG + Sodium Ascorbyl phosphate	CN-LIG+ Lutein 0.5%	CN-LIG	CN-LIG + Niacinamide 5%	CN	CN-LIG + Glycyrrhetinic Acid 0.2%
1:10	0.05	0.1	0.55	0.16	0.67	0.4	1.45 *	0.12
1:20	0.0475	0.096	0.67	0.2	0.7	0.5	1.4	0.33
1:50	0.044	0.115	0.66	0.56	0.77	0.48	1.4	1.02
1:100	0.048	0.5	0.74	0.55	0.7	0.5	1.2	1.05
1:200	0.046	0.52	0.68	0.68	0.75	0.6	1.25	1.25 *
1:500	0.145	0.98	1.2 *	0.99	1.34 *	0.7	0.95	0.99
1:1000	0.173	0.98	1.04	0.92	1.1 ± 5	0.74	0.9	0.95
1:1500	0.7	0.92	0.93	0.98	0.96	0.98 *	0.92	0.93
1:2000	0.77 *	1.13 *	0.87	1.12 *	0.99	0.88	0.87	0.95
1:4000	0.756	0.93	0.9	0.9	0.98	0.92	0.93	0.9

* O.D. is 1.15.

## Data Availability

The data presented in this study are available on request from the corresponding author.

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
