# Peer review of "Chitin Nanofibril-Nanolignin Complexes as Carriers of Functional Molecules for Skin Contact Applications"

_nanomaterials, 2022, doi:10.3390/nano12081295_

Round 1
Reviewer 1 Report
The proposed manuscript describe the application of chitin nanofibrils and nanolignin to embed active molecules such as Vitamin E, Vitamin C, lutein, niacinamide, and glycyrrhetinic acid, in the design of active complexes for skin-contact products. The authors, however, have published at least two manuscripts on the same topic that report the same results highlighted in the proposed manuscript, one in particular on the glycyrrhetinic acid and the other on the Vitamin E. Figures and tables are superimposable. In this form the manuscript cannot be accepted. The authors should modify the superimposable contents and highlight the novelties reported in the new proposed manuscript with respect to those already published.
- Danti et al. Int. J. Mol. Sci. 2019, 20, 2669
- Panariello et al. Cosmetics 2021, 8(2), 27
Author Response
The proposed manuscript describe the application of chitin nanofibrils and nanolignin to embed active molecules such as Vitamin E, Vitamin C, lutein, niacinamide, and glycyrrhetinic acid, in the design of active complexes for skin-contact products. The authors, however, have published at least two manuscripts on the same topic that report the same results highlighted in the proposed manuscript, one in particular on the glycyrrhetinic acid and the other on the Vitamin E. Figures and tables are superimposable. In this form the manuscript cannot be accepted. The authors should modify the superimposable contents and highlight the novelties reported in the new proposed manuscript with respect to those already published.
- Danti et al. Int. J. Mol. Sci. 2019, 20, 2669
- Panariello et al. Cosmetics 2021, 8(2), 27
Thanks to the reviewer for the suggestions, giving us the possibility of improving our paper. The previous papers, that were cited in this paper, are completely different from the one submitted to nanomaterials. In fact, Danti et al. is addressed to more specific biomedical applications and considers only the glycyrrhetinic acid modified complex. Thus, it did not deal with all the others that are present in this paper, where a more systematic and comparative approach is followed.
Panariello et al. paper is addressed at preparing beauty masks, and only the application of the Vitamin E modified complex was considered.
It is thus evident that the present paper represents an important study that completes and integrates the applicative sectorial investigations related to CN-NL complexes. It is significantly different than the previous ones and it presents novel results. In particular:
- A platform of different CN-NL complexes including Vitamin E, Vitamin C, lutein, niacinamide, and glycyrrhetinic acid (derived from liquorice) is presented considering the functional properties of the different molecules (introductory part);
- comparison between the thermal stability, compatibility with keratinocytes between different CN-NL complexes is provided that was not obviously present in the previous papers, where only one complex was used; accordingly, after the comparative analysis of data, the different complexes are compared in Figure 12.
- The effect on nano and micro morphology and thermal stability of the PEG additive, much important for the spray drying process of CNs and CN-NL, was deeply explained. This study was not reported in any other paper and is significant.
The present paper is thus addressed at more general chemical, physical and biological aspects of complexes including nanostructured biomaterials and thus suitable, in terms of topic, for Nanomaterials journal.
Thus, considering chitin nanofibrils, it is also suitable for the specific "Polysaccharides in High-Performance Nanostructured Materials" special issue.
Some sentences were inserted in the manuscript to better highlight this point.
Figure 3, 8 and 9 were replaced with superimposed versions.
Reviewer 2 Report
The manuscript of Coltelli et al., nanomaterials-1598988, “Chitin nanofibrils-nanolignin complexes embedding functional molecules: structure, thermal stability, and in vitro skin compatibility to design new bio-based products for skin contact applications” present a research that could be of interest for the journal’s target readers if important changes are made to improve its values.
The authors should be objective with the results, correctly present all the methods and all the results. Some data seem to be ignored, especially the biological assays. Overall the editing is reckless, making the review process somehow difficult.
The introduction is too long. I recommend to shorten it, for example the paragraph about cytokines (lines 143-159) should be eliminated. It is out of the scope of the research performed.
The section 168-179 is just another abstract of the paper. It should presents the objective of the paper, not a summary of the work performed.
line 101, the authors should use the preferred chemical name of “nicotinamide”, and not niacinamide. Please correct the abstract, figures, and all the paper.
Line 101 and the whole paper: present correctly the references. The information used from the source 59 is only one word, “Niacinamide [59]”? The numbers should reflect the final of the section.
Lines 182 and 197, detail the methods. Remove “P.M.” from the text.
Line 199, it should be vitamin, not “vitamine”. The authors need to check the English editing of the whole paper.
In table 1, and the whole paper: use the decimal point, not comma.
Line 207, in figure 1. Chemically, sodium ascorbyl phosphate is not vitamin C! Vitamin C is L-ascorbic acid. Check and correct all the paper and the figure.
Row 214, use subscript for MgCl2 . See also row 228: cm-1. Line 247. Properly edit the manuscript.
line 223 hMSCs – explain the abbreviation
line 226 – what is MAVI? Please detail on it.
line 236 – ” Hacat cell line”
Line 303, „The shape and intensity of these peaks will change”.. Please comment if in this case, in the authors FTIR experiment the peaks changed. How? Is it a significant change?
Row 327, “spectrum of NL used in this work (not shown)”The authors should present also this IR spectra. They should argue if the spectrum of CN-NL is different or simply a mathematical sum of the individual spectra of CN and NL. This question refers to all the spectra for mixtures presented in the paper.
figure 4 – what is in panel (l)? add the description for the last panel
Figure 6, the structure presented is NOT sodium ascorbyl phosphate. In the figure’s legend explain what VC, NI and LU are referring to.
The authors should use a more compact format for table 5. It is very hard to understand. The table could be presented with several columns for each substance and several rows for the corresponding concentration.
On line 503, the OD for the control is 4. In my opinion is a very high value. I wouldn’t trust a OD over 1. The authors should check if the Beer-Lambert law can be still applied. It could be a major source of error. The authors should address this problem by doing again the experiment. The OD values are not mathematically connected with the concentration. See for example CN-LIG. It looks quite random.
Another problem is that the MTT assay results are not interpreted. What does all mean? Some substances have a very important cytotoxic effect. The authors should comment on this.
Figure 11 (it should be 10) has a very low quality. It should be prepared again as to really understand it. The authors just presented their results without almost any comment on them. Are the results significant? Add a significance evaluation before drawing any conclusions.
The authors need to present the anti-inflammatory and immune responses assay results in correlation with their cytotoxic effects.
Line 541, figure 19?
Line 541, the term biocompatibility is not the best choice. The authors performed only a MTT test. There should be more tests performed before declaring the materials biocompatible, like for example irritation and sensitization tests.
The authors should correct the references. There is a very high number of references that are only added to artificially enhance the scientometric data of the authors. Just one example, reference 1 is an editorial on a project. How is this relevant?
Next, the authors add together 12 references for only 2 lines of writing (line 43) and another 12 for the next 3 lines (row 46). Select only the relevant articles and present them in such a way that any information could be traced back to the citing article. All selfcitations should be removed if not clearly relevant.
Author Response
The manuscript of Coltelli et al., nanomaterials-1598988, “Chitin nanofibrils-nanolignin complexes embedding functional molecules: structure, thermal stability, and in vitro skin compatibility to design new bio-based products for skin contact applications” present a research that could be of interest for the journal’s target readers if important changes are made to improve its values.
The authors should be objective with the results, correctly present all the methods and all the results. Some data seem to be ignored, especially the biological assays. Overall the editing is reckless, making the review process somehow difficult.
The manuscript was revised following all the indications of the reviewer
The introduction is too long. I recommend to shorten it, for example the paragraph about cytokines (lines 143-159) should be eliminated. It is out of the scope of the research performed.
The introductory part was shortened. The part related to cytokines was removed.
The section 168-179 is just another abstract of the paper. It should presents the objective of the paper, not a summary of the work performed.
This part was summarized, evidencing mainly the objectives
line 101, the authors should use the preferred chemical name of “nicotinamide”, and not niacinamide. Please correct the abstract, figures, and all the paper.
This change was done along all the manuscript
Line 101 and the whole paper: present correctly the references. The information used from the source 59 is only one word, “Niacinamide [59]”? The numbers should reflect the final of the section.
The number was shifted at the end of the sentence. This reference is a general review regarding nicotinamide.
Lines 182 and 197, detail the methods. Remove “P.M.” from the text.
Yes, sorry for having forgotten this annotation in the text.
Line 199, it should be vitamin, not “vitamine”. The authors need to check the English editing of the whole paper.
Yes, this change was done
In table 1, and the whole paper: use the decimal point, not comma.
Yes, this is checked
Line 207, in figure 1. Chemically, sodium ascorbyl phosphate is not vitamin C! Vitamin C is L-ascorbic acid. Check and correct all the paper and the figure.
The manuscript was fully revised
Row 214, use subscript for MgCl2 . See also row 228: cm-1. Line 247. Properly edit the manuscript.
Yes, the manuscript was fully edited
line 223 hMSCs – explain the abbreviation
This is an error. It was removed
line 226 – what is MAVI? Please detail on it.
It is a mistake. It is removed.
line 236 – ” Hacat cell line”
HaCaT is a spontaneously transformed aneuploid immortal keratinocyte cell line from adult human skin widely used in scientific research. The word Hacat was put in brackets
Line 303, „The shape and intensity of these peaks will change”.. Please comment if in this case, in the authors FTIR experiment the peaks changed. How? Is it a significant change?
In our case not significant changes were observed.
Row 327, “spectrum of NL used in this work (not shown)”The authors should present also this IR spectra. They should argue if the spectrum of CN-NL is different or simply a mathematical sum of the individual spectra of CN and NL. This question refers to all the spectra for mixtures presented in the paper.
The spectrum of NL was added to Figure 2 and a sentence was also added in the manuscript, evidencing that the slight shift of peaks is attributable to the interactions between CN and NL.
figure 4 – what is in panel (l)? add the description for the last panel
Yes, the (l) panel was added. Thanks.
Figure 6, the structure presented is NOT sodium ascorbyl phosphate. In the figure’s legend explain what VC, NI and LU are referring to.
The formula of sodium ascorbyl phosphate was revised both in figure 1 and 6. Then, the abbreviations were inserted in Figure 1 for a better clarification
The authors should use a more compact format for table 5. It is very hard to understand. The table could be presented with several columns for each substance and several rows for the corresponding concentration.
On line 503, the OD for the control is 4. In my opinion is a very high value. I wouldn’t trust a OD over 1. The authors should check if the Beer-Lambert law can be still applied. It could be a major source of error. The authors should address this problem by doing again the experiment. The OD values are not mathematically connected with the concentration. See for example CN-LIG. It looks quite random.
As rightly requested, the table's formatting has been changed and made more understandable. Furthermore, to avoid running into problems caused by the high absorbance values detected, probably due to the setting of the spectrophotometer, we decided to express the data in terms of Percentage of viable cells following the various treatments compared to untreated cells.
As regards the non-linearity between some values with respect to the dilutions tested, it should be emphasized that in biological systems it can often happen that, after having exceeded certain threshold values, the relationship between activity and concentration of some substances becomes inversely proportional, i.e. that they are more active at higher concentrations, which also affects their cytotoxicity. (Peper A. Aspects of the Relationship between Drug Dose and Drug Effect. Dose-Response. April 2009)
Another problem is that the MTT assay results are not interpreted. What does all mean? Some substances have a very important cytotoxic effect. The authors should comment on this.
As specified in the Materials and Methods section (lines 219-246) and in the Results section (lines 473-483), the MTT assay was conducted in order to establish what was the optimal and not-cytotoxic concentration for each substance, in order to be used in subsequent experiments. However, a description has been added in paragraph 3.3 in the Results section in which the data obtained are commented and the concentrations chosen for each substance are indicated.
Figure 11 (it should be 10) has a very low quality. It should be prepared again as to really understand it. The authors just presented their results without almost any comment on them. Are the results significant? Add a significance evaluation before drawing any conclusions.
The quality of the graphs has been improved and statistical analysis with p-value has also been added. A description of the obtained findings at different time points and their meaning was given.
The authors need to present the anti-inflammatory and immune responses assay results in correlation with their cytotoxic effects.
We cannot present the data as requested as the evaluation of the induction of the immune response was made on not-cytotoxic concentrations, as at cytotoxic concentrations most of the cells die and we are unable to obtain sufficient quantities of mRNA to perform the Real-Time PCR.
Line 541, figure 19?
Figure 19 was replaced with a figure with a higher resolution
Line 541, the term biocompatibility is not the best choice. The authors performed only a MTT test. There should be more tests performed before declaring the materials biocompatible, like for example irritation and sensitization tests.
As a rule, skin sensitization tests are performed on finished products, this was only an initial screening on raw materials to establish their working concentrations. We use the term biocompatibility broadly referring not only to the response to the MTT cytotoxicity assay, but also to the ability to stimulate or not an inflammatory response in the treated cells, given that it was evaluated by Real Time PCR.
The authors should correct the references. There is a very high number of references that are only added to artificially enhance the scientometric data of the authors. Just one example, reference 1 is an editorial on a project. How is this relevant?
Reference 1 was removed. It was an introduction to a book related to use of biobased materials in applications in contact with skin, so it was pertinent. However, as other papers are also cited, it could be removed.
Next, the authors add together 12 references for only 2 lines of writing (line 43) and another 12 for the next 3 lines (row 46). Select only the relevant articles and present them in such a way that any information could be traced back to the citing article. All selfcitations should be removed if not clearly relevant.
10 references, considered less representative or partially written by some of the co-authors of this paper, were removed.
Thanks for the useful suggestions
Reviewer 3 Report
The reviewed manuscript has practical relevance and appears to present meaningful results worthy of publication. Nevertheless, it is impossible for the reviewer to assess the manuscript in its current form, as the structure is very confusing and results are presented in a manner that does not allow for their interpretation.
The manuscript needs to undergo significant changes, and be re-evaluated once the authors address several important points, which in my opinion require rectification/clarification.
- There are numerous instances of run-on sentences and random comma placement, often confusing the primary points of the sentence (e.g. lines 21-24). Interestingly, there are also multiple sentences with an evident lack of punctuation (e.g. lines 41-43)!
- Abbreviations should be avoided in the Abstract, especially in cases where the abbreviated term is not re-used in the abstract (see “TGA” line 29).
- Abbreviations should also be only placed once. Chitin nanofibrils for example are abbreviated multiple times (line 21, line 47, line 563, line 569 and line 576).
- The abbreviation M. appears for the first time in line 182, but is not explained anywhere in the text… does this refer to one of the manuscript’s authors? Please elaborate.
- The authors use an excessive amount of references for the type of article (Research). I think these should be reduced to half, e.g. page1 lines 41-43, there is no need for 12 references for this particular statement!
- The introduction is too long, at some points over-analyzing points that are largely considered as basic knowledge for a scientific audience.
- Chapter numbers are very confusing, Materials and Methods are numbers as chapter 2 but sub-chapters are numbered 4.1, 4.2 and so on, whereas chapter 4 is missing altogether!
- Page 1, line 32, I think “dilutions” is not an appropriate term here. Maybe concentration or something else would be better suited.
- Figure legends are mostly impossible to read (e.g. figure 8). Please use a picture manipulating software to adjust the font size within the figures. It is not possible to assess the presented results otherwise.
There are multiple other syntactical and linguistic issues throughout the text and I’d urge the authors to carefully proof-reed their manuscript prior to re-submitting it.
The Materials and Methods section is in many parts deficient in details. Although the authors cite their previous work, they need to provide a brief description along this. For example Silver Nanoparticles are mentioned in line 197, but no information is provided as to their physicochemical characteristics (shape, size, morphology or even concentration), unless I missed this due to the structure of the manuscript.
Author Response
The reviewed manuscript has practical relevance and appears to present meaningful results worthy of publication. Nevertheless, it is impossible for the reviewer to assess the manuscript in its current form, as the structure is very confusing and results are presented in a manner that does not allow for their interpretation.
The manuscript needs to undergo significant changes, and be re-evaluated once the authors address several important points, which in my opinion require rectification/clarification.
- There are numerous instances of run-on sentences and random comma placement, often confusing the primary points of the sentence (e.g. lines 21-24). Interestingly, there are also multiple sentences with an evident lack of punctuation (e.g. lines 41-43)!
Punctuation was controlled and revised.
- Abbreviations should be avoided in the Abstract, especially in cases where the abbreviated term is not re-used in the abstract (see “TGA” line 29).
Abbreviations were revised in the abstract
- Abbreviations should also be only placed once. Chitin nanofibrils for example are abbreviated multiple times (line 21, line 47, line 563, line 569 and line 576).
Thank you, these lines were revised accordingly.
- The abbreviation M. appears for the first time in line 182, but is not explained anywhere in the text… does this refer to one of the manuscript’s authors? Please elaborate.
No, this acronym was never used before. The definition is reported in the introductory part, for making clearer the paper. Moreover, M is used instead of m.
- The authors use an excessive amount of references for the type of article (Research). I think these should be reduced to half, e.g. page1 lines 41-43, there is no need for 12 references for this particular statement!
10 references, considered less representative or partially written by some of the co-authors of this paper, were removed in the introductory part.
Anyway, the high number of references is not only due to the introductory part, but also to the necessity of citing papers during the discussion of results. These references are considered important for the quality of the paper, so they were maintained in the bibliography.
- The introduction is too long, at some points over-analyzing points that are largely considered as basic knowledge for a scientific audience.
The introductory part was reduced, removing some inessential information and also removing the part related to biocompatibility tests that resulted too much extensive. The last part was limited at describing the paper objectives.
- Chapter numbers are very confusing, Materials and Methods are numbers as chapter 2 but sub-chapters are numbered 4.1, 4.2 and so on, whereas chapter 4 is missing altogether!
Thank you and we apologize for this error. The numbers were revised.
- Page 1, line 32, I think “dilutions” is not an appropriate term here. Maybe concentration or something else would be better suited.
The word was replaced with “concentrations”.
- Figure legends are mostly impossible to read (e.g. figure 8). Please use a picture manipulating software to adjust the font size within the figures. It is not possible to assess the presented results otherwise.
Figure 3,8 and 9 were replaced with versions in which superposition of curves was preferred, to avoid the presence of small unreadable words. For figure 11 the resolution was increased.
There are multiple other syntactical and linguistic issues throughout the text and I’d urge the authors to carefully proof-reed their manuscript prior to re-submitting it.
The text was fully revised on this point of view. We apologize.
The Materials and Methods section is in many parts deficient in details. Although the authors cite their previous work, they need to provide a brief description along this. For example Silver Nanoparticles are mentioned in line 197, but no information is provided as to their physicochemical characteristics (shape, size, morphology or even concentration), unless I missed this due to the structure of the manuscript.
In the paper silver nanoparticles were considered as a “commercial” reference. Their characterization was performed in a previous paper that is added in manuscript references. A narrow size distribution with an average size equal to 61,9 nm was measured for silver nanoparticles. The paper is: Antibacterial and Anti-inflammatory Green Nanocomposites by Pierfrancesco Morganti, Paola Del Ciotto, Marco Stoller, Angelo Chianese. It is cited as reference 63.
Thank you for the useful comments and suggestions.
Reviewer 4 Report
The paper problematic’s deals with encapsulated bioactive compounds that may handle skin aging through exposure to mainly antioxydative processes. The experiments are well performed and rather consistent for analytic purposes.
The major issues concern the Results section with both graphs and tables that are displayed in a very confusing manner, hard to read (chemical structures) or unhandy (Fig. 11 and Table 5 specially). Besides most of them should be supplied as supplementary data. One questions also why 24 hours data are presented whereas only 6hours are discussed.
More, the spelling of many words must be improved (Mm, CN-NL and CN-NL and so many others)
As a whole, the reviewer recommends major revisions to amend the paper in order to be much more concise and avoid a loosy data presentation.
Author Response
The paper problematic’s deals with encapsulated bioactive compounds that may handle skin aging through exposure to mainly antioxydative processes. The experiments are well performed and rather consistent for analytic purposes.
The major issues concern the Results section with both graphs and tables that are displayed in a very confusing manner, hard to read (chemical structures) or unhandy (Fig. 11 and Table 5 specially). Besides most of them should be supplied as supplementary data. One questions also why 24 hours data are presented whereas only 6hours are discussed.
Figure 11 and table 5 (now Tbale 6) were replaced, improving the discussion of this part of the paper. Figures and scheme related to chemical structures were revised. Figure 3, 8 and 9 were replaced with superimposed versions. We have now added and discussed both time points, 6 h and 24 h.
More, the spelling of many words must be improved (Mm, CN-NL and CN-NL and so many others)
As a whole, the reviewer recommends major revisions to amend the paper in order to be much more concise and avoid a loosy data presentation.
The acronyms and abbreviations were revised to make them clearer. The introductory part was reduced and better focused on objectives.
Thanks for the useful suggestions
Round 2
Reviewer 1 Report
In this form the manuscript can be accepted.
Author Response
Dear Reviewer,
thank you for your usefull suggestions. We are glad that our current version can be accepted.
Regards
Beatrice
Reviewer 2 Report
The authors made some changes to their manuscript, but several important elements were ignored or the changes or very little. The authors' response is in many instances short and in some case unconvincing.
See as example, „Line 303, The shape and intensity of these peaks will change”.. Please comment if in this case, in the authors FTIR experiment the peaks changed. How? Is it a significant change?”. The answers should be presented in the paper, not only for the reviewer.
The preferred chemical name of “nicotinamide”, and not niacinamide. Please correct the figure.
Figure 6. In the figure’s legend explain what VC, NI and LU are referring to.
I suggest the authors to check again all the comment of the first review and check if their response is also in the paper, not only for the reviewer.
In the first review I wrote “On line 503, the OD for the control is 4. In my opinion is a very high value. I wouldn’t trust a OD over 1. The authors should check if the Beer-Lambert law can be still applied. It could be a major source of error. The authors should address this problem by doing again the experiment. The OD values are not mathematically connected with the concentration. See for example CN-LIG. It looks quite random”.
The response “As rightly requested, the table's formatting has been changed and made more understandable. Furthermore, to avoid running into problems caused by the high absorbance values detected, probably due to the setting of the spectrophotometer, we decided to express the data in terms of Percentage of viable cells following the various treatments compared to untreated cells. ” ignores the real problem. We cannot use the Beer-Lambert law because OD values are very high and therefore we don’t have a correct representation of the number of cells. The authors must perform again the experiment. There is no way around it. The second response is out of the context “As regards the non-linearity between some values with respect to the dilutions tested, it should be emphasized that in biological systems it can often happen that, after having exceeded certain threshold values, the relationship between activity and concentration of some substances becomes inversely proportional, i.e. that they are more active at higher concentrations, which also affects their cytotoxicity”. In my opinion the author make another mistake: “Viability values above 100%, namely, an increment in cell viability, was obtained…”. Biologically speaking there is no viability over 100%, but it may be present a proliferative effect. The authors should correctly interpret the data and discuss them. It seems that some of the new materials have a proliferative effect at small doses that could be a safety problem. Please discuss this aspect in the paper.
Author Response
The authors made some changes to their manuscript, but several important elements were ignored or the changes or very little. The authors' response is in many instances short and in some case unconvincing.
See as example, „Line 303, The shape and intensity of these peaks will change”.. Please comment if in this case, in the authors FTIR experiment the peaks changed. How? Is it a significant change?”. The answers should be presented in the paper, not only for the reviewer.
We have added the sentence : “Peaks at 3100-3440 cm-1, influenced by hydrogen bonds, were not significantly altered by the presence of PEG and NL” at line 343.
The preferred chemical name of “nicotinamide”, and not niacinamide. Please correct the figure.
Ok, sorry, the modification of this figure was forgotten in our previous revision. We apologize. Now it is ok.
Figure 6. In the figure’s legend explain what VC, NI and LU are referring to.
Figure 1 and Table 1 are reporting all the abbreviations adopted in the manuscript. For this reason, we believed that the legend could be clear. However, we also have added the abbreviations in the figure 5 and 6. Thanks.
I suggest the authors to check again all the comment of the first review and check if their response is also in the paper, not only for the reviewer.
Yes, it was done and all the requests were effectively considered in the manuscript now. Thanks. We apologize for some lapses.
In the first review I wrote “On line 503, the OD for the control is 4. In my opinion is a very high value. I wouldn’t trust a OD over 1. The authors should check if the Beer-Lambert law can be still applied. It could be a major source of error. The authors should address this problem by doing again the experiment. The OD values are not mathematically connected with the concentration. See for example CN-LIG. It looks quite random”.
The response “As rightly requested, the table's formatting has been changed and made more understandable. Furthermore, to avoid running into problems caused by the high absorbance values detected, probably due to the setting of the spectrophotometer, we decided to express the data in terms of Percentage of viable cells following the various treatments compared to untreated cells. ” ignores the real problem. We cannot use the Beer-Lambert law because OD values are very high and therefore we don’t have a correct representation of the number of cells. The authors must perform again the experiment. There is no way around it.
As requested, the experiment was repeated, by optimizing cell density and appropriately diluting the supernatants to obtain suitable and reproducible O.D. values. Some discrepancies have been reduced, even if the trend is confirmed.
|
CN-LIG+ AgNPs |
CN-LIG+Vit.E 0,2% |
CN-LIG+Sodium Ascorbyl phosphate |
CN-LIG+ Lutein 0,5% |
CN-LIG |
CN-LIG+ Niacinamide 5% |
CN |
CN-LIG+Glycyrrhetinic acid 0,2% |
|
|
1:10 |
0.05 |
0.1 |
0.55 |
0.16 |
0.67 |
0.4 |
1.45* |
0.12 |
|
1:20 |
0.0475 |
0.096 |
0.67 |
0.2 |
0.7 |
0.5 |
1.4 |
0.33 |
|
1:50 |
0.044 |
0.115 |
0.66 |
0.56 |
0.77 |
0.48 |
1.4 |
1.02 |
|
1:100 |
0.048 |
0.5 |
0.74 |
0.55 |
0.7 |
0.5 |
1.2 |
1.05 |
|
1:200 |
0.046 |
0.52 |
0.68 |
0.68 |
0.75 |
0.6 |
1.25 |
1.25* |
|
1:500 |
0.145 |
0.98 |
1.2* |
0.99 |
1.34* |
0.7 |
0.95 |
0.99 |
|
1:1000 |
0.173 |
0.98 |
1.04 |
0.92 |
1.1±5 |
0.74 |
0.9 |
0.95 |
|
1:1500 |
0.7 |
0.92 |
0.93 |
0.98 |
0.96 |
0.98* |
0.92 |
0.93 |
|
1:2000 |
0.77* |
1.13* |
0.87 |
1.12* |
0.99 |
0.88 |
0.87 |
0.95 |
|
1:4000 |
0.756 |
0.93 |
0.9 |
0.9 |
0.98 |
0.92 |
0.93 |
0.9 |
The second response is out of the context “As regards the non-linearity between some values with respect to the dilutions tested, it should be emphasized that in biological systems it can often happen that, after having exceeded certain threshold values, the relationship between activity and concentration of some substances becomes inversely proportional, i.e. that they are more active at higher concentrations, which also affects their cytotoxicity”. In my opinion the author make another mistake: “Viability values above 100%, namely, an increment in cell viability, was obtained…”. Biologically speaking there is no viability over 100%, but it may be present a proliferative effect. The authors should correctly interpret the data and discuss them. It seems that some of the new materials have a proliferative effect at small doses that could be a safety problem. Please discuss this aspect in the paper.
As requested, a sentence on the proliferative activity of some compounds has been added at the end of the Results and Discussion section. However, since in the new version of the Tab the data are shown in OD, we want to clarify that the greatest increase in terms of percentage, therefore the greatest proliferative activity, was found mainly in the CN, CN-NL samples and in presence of glycyrrhetinic acid. As already previously indicated in the Results and Discussion section, these compounds have been selected for wound dressing applications, where proliferative activity is definitely required. In addition, these compounds were tested on keratinocytes in subsequent experiments where the immunomodulatory activity was evaluated, and the results obtained are completely comparable to those of the other compounds, (CN even has anti-inflammatory activity) did not show a cellular stress. Moreover, the percentage increase in cellular viability does not reach values that can be defined as "dangerous", and is easily reversible simply by lowering the concentrations of active ingredients.
In all other samples, the CN-NL complexes are associated with natural active compounds that reduce their proliferative power and in fact they have been chosen for other types of applications.
Reviewer 3 Report
The authors have addressed most of my concerns and I believe now the manuscript can be published in its current form
Author Response
Dear Reviewer, thanks for appreciating our revisions and for helping us in improving our work
Regards
Beatrice
Reviewer 4 Report
In the abstract, the abreviations have not to be precised here.
Few gramaatical errors in this paper have been amended, :
line 33: "a of"
line 45: the coma is not well placed
line 50: acetyl-D-glucosamine
lines 79 and 90: as an antioxidant
line 103: enhances DNA repair
line 105: "for example" can be removed
line 124: such as in polymer
line 184: material samples
line 303: the band
line 326: remove "probably"
line 363: you can replace "resulting to be " by forming
line 563: should be done while keeping
Author Response
Dear reviewer,
thank you very much for your kind suggestions. We apologize for some errors still present in the manuscript after our revision. We have done all the suggested revisions.
Few gramaatical errors in this paper have been amended, :
line 33: "a of"
done
line 45: the coma is not well placed
done
line 50: acetyl-D-glucosamine
done
lines 79 and 90: as an antioxidant
done
line 103: enhances DNA repair
done
line 105: "for example" can be removed
done
line 124: such as in polymer
done
line 184: material samples
done
line 303: the band
done
line 326: remove "probably"
done
line 363: you can replace "resulting to be " by forming
done
line 563: should be done while keeping
done
Thank you very much